# Forecasting the Preparatory Phase of Induced Earthquakes by Recurrent Neural Network

**Matteo Picozzi \*** and **Antonio Giovanni Iaccarino**

Department of Physics, University of Naples Federico II, Complesso Universitatio Monte S. Angelo, via Cinthia, 80126 Naples, Italy; antoniogiovanni.iaccarino@unina.it
* Correspondence: matteo.picozzi@unina.it

**Abstract:** Earthquakes prediction is considered the holy grail of seismology. After almost a century of efforts without convincing results, the recent raise of machine learning (ML) methods in conjunction with the deployment of dense seismic networks has boosted new hope in this field. Even if large earthquakes still occur unanticipated, recent laboratory, field, and theoretical studies support the existence of a preparatory phase preceding earthquakes, where small and stable ruptures progressively develop into an unstable and confined zone around the future hypocenter. The problem of recognizing the preparatory phase of earthquakes is of critical importance for mitigating seismic risk for both natural and induced events. Here, we focus on the induced seismicity at The Geysers geothermal field in California. We address the preparatory phase of M~4 earthquakes identification problem by developing a ML approach based on features computed from catalogues, which are used to train a recurrent neural network (RNN). We show that RNN successfully reveal the preparation of M~4 earthquakes. These results confirm the potential of monitoring induced microseismicity and should encourage new research also in predictability of natural earthquakes.

**Keywords:** preparatory phase; earthquake forecasting; induced seismicity





## 1. Introduction

The physics of earthquake initiation is a challenging research field with severe implications for modern society in terms of earthquake forecasting and seismic risk mitigation.

Pioneering studies [1–7] have shown that large magnitude earthquakes can be anticipated by foreshocks and slip instabilities. However, the non-systematic foreshock appearance and the lack of systematic precursory patters led the background physical processes generating foreshocks and the preparatory phase being not fully understood and matter of debate [8,9].

Two main contrasting models have been proposed concerning foreshocks generations. According to some authors [9–11], foreshocks are related to a tectonic loading process associated to aseismic slip, which represents a deviation from the normal behavior of seismicity [12]. This model would support the existence of a preparatory phase for large earthquakes, leaving us with the hope that in future earthquakes will be predictable.

By contrast, for other authors [13,14] foreshocks result from a triggering process that is part of the normal behavior of seismicity (i.e., following the concept of Self-Organized Criticality), for which events can cascade into a larger one, yet without any clear background driving process. The key practical aspect of this model is that the probability of a large event in a short period (e.g., one week) following to the occurrence of foreshocks is very low, and therefore of limited use [12].

Over the last decade, significant advances have been obtained in this research field thanks to the availability of high-resolution seismic catalogs, which resulted from efforts done by the seismological community in increasing the number of dense seismic networks deployed nearby active faults and in the development of advanced event detection techniques. A recent study [15] has shown, indeed, that thank to enriched catalogues foreshock

activity in California is much more common than previously understood, with foreshocks that have been found in 72% of cases for a sample of 46 mainshocks.

However, it must keep in mind that "foreshocks" is a label assigned to earthquakes retrospectively. Instead of trying to single out if an event is a foreshock or not, a promising route seems observing the collective spatial and temporal behavior of small magnitude earthquakes for unveiling if seismic or aseismic preparatory processes are underway.

As result of the scientific developments in infrastructures and data-mining strategies, systematic patterns in seismicity preceding large earthquakes has started to emerge [16–25], showing that micro and small magnitude events before large earthquake show temporal and spatial peculiar patterns in their evolution. These observations support the idea that, at least in some cases, the interpretative model for which foreshocks are related to a tectonic loading process associated to aseismic slip [9–11] is reliable.

A recent, systematic review on the initiation of large earthquakes [26] has highlighted that their generation is the result of complex, multiscale processes where the tectonic environment and external factors (e.g., natural and/or anthropogenic inputs that impact on the local stress-field) interact. The resultant integrated generation model [26] proposes a progressive localization of shear deformation around a rupture zone, which evolves into a final rapid loading (i.e., generating foreshocks) of a localized volume nearby the hypocenter of the major dynamic rupture. Such kind of process might be universal. Indeed, similar patterns of damage evolution across a fault zone have been found also studying the temporal and spatial distribution and characteristics of acoustic emissions during triaxial tests on rock samples [27].

We outline the generation processes of large earthquakes model in Figure 1. Here, we summarize the spatio-temporal evolution of small events before and after a large earthquake together with the expected seismic parameters trend (i.e., see below for their explanation), which have been identified as good indicators for the damage evolution (see [26,27] and the references therein).

In view of these studies [26,27], monitoring the spatio-temporal evolution of micro-seismicity could lead recognizing the preparatory phase of moderate to large earthquakes a reachable goal.

A meta-analysis of seismic sequences preceding large earthquakes [12] has highlighted that when observed in nature, the preparatory phase is potentially identifiable if the detected seismic events are more than three orders of magnitude lower that the mainshock. This justify why the preparatory phase is generally difficult to detect and observe for moderate size earthquake, unless enriched catalogues are available [28,29].

Recent works explored the use of machine learning (ML) to predict the time remaining before rupture in laboratory experiments [30], the spatial distribution of aftershocks [31], or the magnitude [32].

We propose here, for the first time, the use of recurrent neural networks (RNN) to capture the preparatory phase of moderate earthquakes by studying the spatio-temporal evolution of micro-seismicity.

To this aim, we focus on data relevant to The Geysers geothermal field in California (Figure 2). The crustal volume below The Geysers can be seen as a complex physical system whose stress field evolves due to the interaction of tectonic processes and industrial operations in a chaotic way. Our aim is to verify if larger events are anticipated by transients in features sensitive to the evolution of the crustal damage process.

The Geysers hosts a high quality, dense seismic network to monitor the high seismicity rate since 2003 [33]. Considering the period between 2003 and 2016, indeed, the official catalogue (NCEDC, https://ncedc.org/egs) includes ~460.000 events in the magnitude range between M −0.7 and M 4.3 (Figure 2a). Within this dataset, we have identified eight events with moment magnitude larger than $M_w$ 3.9 (hereinafter, M4 events) occurred between 2006 and 2014 within the geothermal field (Figure 2a,d), and we have estimated the completeness magnitude, $M_c$, being equal to M 0.5 (Figure 1c), which, according to Mignan (2014) [12], makes us hopeful to observe their preparatory phase. In this work, the

available local magnitude, $M_L$, estimates available in the NCEDC seismic catalogue have converted in moment magnitude scale, $M_w$ [34]; (see section Materials and Methods for more details).

The seismic catalogue has been analyzed to extract nine features describing different aspects of the temporal and spatial evolution of seismicity. The considered features are: the *b-value* and completeness magnitude, $M_c$, of the Gutenberg–Richter law [35]; the fractal dimension of hypocenters, $D_c$ [36]; the generalized distance between pairs of earthquakes, $\eta$ [37]; the Shannon's information entropy, $h$ [38–40]; the moment magnitude, $M_w$, and moment rate, $\dot{M}_0$; the total duration of event groups, $\Delta T$, and the inter-event time, $\Delta t$ (see section Materials and Methods for more details about the seismic features).

The crustal volume below The Geysers can be seen as a complex physical system which stress field evolves due to the interaction of tectonic processes and industrial operations in a chaotic way. In similar conditions, our aim is to verify if larger events are anticipated by transients in features sensitive to the evolution of the crustal damage process.

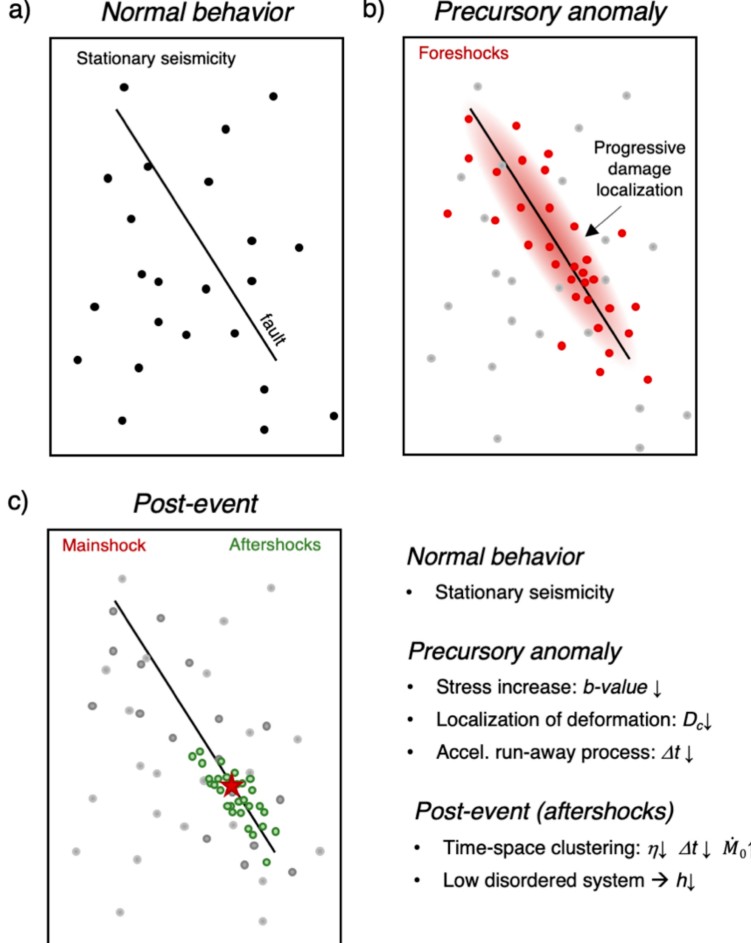

**Figure 1.** Schematic illustration of the processes anticipating and following a large earthquake. (**a**) Normal behavior phase, during which the seismicity shows a spatio-temporal stationary nature. (**b**) Precursory anomaly phase, during which the localization of deformation proposed by Kato and Ben-Zion, 2020 [26], occurs. Seismicity can show *b-value*, fractal dimension ($D_c$), and seismic-rate ($\Delta t$) decrease. (**c**) Post-event phase, during which the seismicity is characterized by time-space clustering (i.e., $\Delta t$ and generalized-distance, $\eta$, decrease, while moment-rate, $\dot{M}_0$, increase) and the system shows high order (Shannon entropy, $h$, decrease).

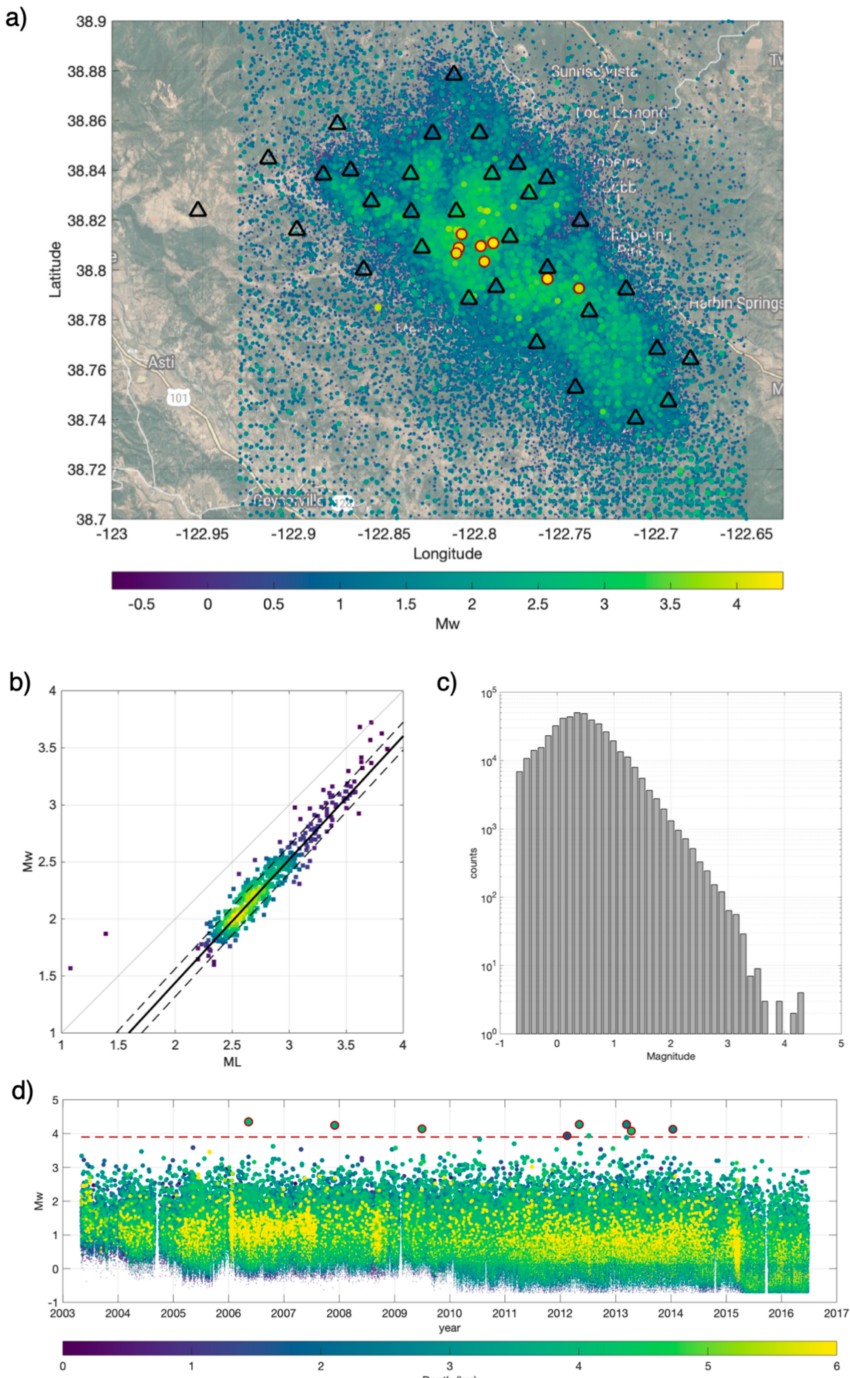

**Figure 2.** (**a**) Locations of the earthquakes considered in this study (data are colored and have size varying per $M_w$) and seismic stations of the Berkeley–Geysers, BG code, seismic network (black triangles). M4 earthquakes are identified by a red contour. The map was made using MATLAB (R2019b; https://it.mathworks.com/, last accessed October 2020). (**b**) Local magnitude, $M_L$, versus moment magnitude, $M_w$ scatter plot (viridis color scale with yellow for higher density of data). The trend lines defined by linear regression is shown in black (thick black line) along with their $\pm 1$ standard deviation (upper and lower dashed black lines). The 1:1 scaling line is also reported (thin black line). (**c**) Histogram showing the distribution of magnitude for the considered events. (**d**) Temporal distribution of magnitude (data are colored and have size varying per hypocentral depth). M4 earthquakes are identified by a red contour. M3.9 threshold (red dashed line).

We look for transients with respect to the stationary occurrence in seismicity that could hints for the preparation process of larger earthquakes. So, besides the typical features related to the rate, inter-event time, and seismic moment rate of events, we have explored the use of features carrying information on the stress condition of the crust and on the evolving characteristics of earthquake patterns.

Laboratory studies [41,42] have shown that variations in the frequency-magnitude earthquakes distribution, expressed in terms of the *b-value* of the Gutenberg–Richter law [35], were connected to higher density of fractures and moment release. Recently, it has been shown [43] that transients in *b-value* in aftershock zones can be used to foresee if after a large event an even stronger one yet to come. These observations are related to the connection existing between *b-value* and the stress conditions associated to ruptures [44,45].

A further point of view in studying microseismicity is provided by the fractal dimension of hypocenters [36], $D_c$, which has been reported to vary before large earthquakes [46–48]. As discussed by many authors on both theoretical and experimental basis [49–54], *b-value* and $D_c$ are strongly correlated, especially when the self-similar scaling assumption for earthquakes holds. Following the computation of *b-value* and $D_c$, we have estimated the generalized distance between pairs of earthquakes [37], $\eta$, which includes the contribution of both of them.

Finally, we have quantified the level of organization of the earthquakes population in our complex dynamic system by the Shannon's information entropy [38–40].

The sequential data (features) have been analyzed by the gated recurrent units (GRU) [55], an advanced RNN with strong capability of analyzing data non-linearly correlated and characterized by long-term trends that classic RNN algorithms are not able to handle. Throughout the last years, applications of similar algorithms (i.e., the Long Short-Term Memory, LSTM [56]) and other neural network [57,58] for earthquake prediction have been proposed, but the literature survey proposed by Mignan and Broccardo (2020) [59] led the authors to conclude that "the potential of deep learning to improve earthquake forecasting remains unproven".

In our work, we explored if RNN can detect and exploit the complex pattern of different potential seismic precursors to unveil the preparatory processes of larger magnitude induced earthquakes.

Our sensitivity analysis highlighted that among the selected features, some are more efficient in detecting the preparatory phase, while others performed better for the automatic identification of aftershocks. For these reasons, we trained two RNNs on five series of hundreds to thousands of earthquakes around M4 events occurred at The Geysers in the period 2006–2012.

Our cutting-edge outcome, from the application of the RNNs to three earthquakes series occurred in the same area after 2013 and including M4 events, is that for all of them several hours of preparatory phase can be automatically identified. Our results demonstrate that machine learning approaches have high potential of catching complex pattern in seismic features and assisting operators in the induced seismicity management.

The monitoring of induced seismicity is a peculiar context, where the availability of high quality and dense seismic networks lead to enriched catalogues that in turns can allow high quality training of artificial intelligence approaches. However, we believe our study will also stimulate other researchers to explore similar multi-parameters machine-learning approaches to areas where natural large magnitude earthquakes occur.

## 2. Features

### 2.1. Features Computation and General Working Outline

Information extracted from the seismic catalogue (i.e., hypocenter coordinates, local magnitude, and occurrence date and time) are used to compute the series of features used by RNNs.

We followed the same analysis scheme for all features (Figure 3a) but for $M_w$ and $\Delta t$ that are computed for each single event. That is to say, the features are computed on

windows of *n* events (with *n* = 200) and their value is assigned to the *n*th event at the time $t_i$ and windows move of one event at time (i.e., each feature represents a time series). Here, *n* equal to 200 was selected considering that this parameter is important for obtaining robust estimates of the *b-value* [60]. For instance, Dresen et al., 2020 [27], used *n* equal to 150.

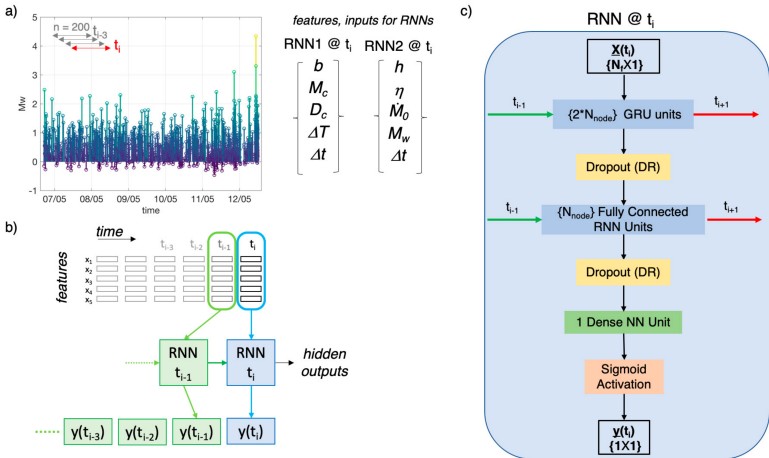

**Figure 3.** Overview of the recurrent neural network (RNN)-based preparatory phase prediction. (**a**) Computation of features on moving windows of *n* events. (**b**) Outline of the RNNs algorithms. (**c**) Outline of the RNN architecture at time $t_i$.

In the following, we detail the computation of features (Sections 2.2–2.7), while we describe in Section 3 the RNN algorithm and the step-by-step procedure implemented in our study.

### 2.2. Duration of Event Groups and Inter-Event Time

We compute for each windows of *n* events the total duration of event groups ($\Delta T = t_n \eta t_{1st}$) and the inter-event time ($\Delta t = t_n - t_{n-1}$).

### 2.3. Moment Magnitude and Moment Rate

We converted the local magnitude ($M_L$) in moment magnitude scale ($M_w$) benefiting from the characterization of around six hundred earthquakes at The Geysers [61] (i.e., characterized in terms of seismic moment, $M_0$, and corner frequency, Figure 2b). The $M_0$ values have been thus converted in $M_w$, and we parameterized the following $M_L$ vs. $M_w$ relationship:

$$M_w = 1.08 \times M_L - 0.72, \qquad \sigma = 0.12, \tag{1}$$

The moment rate, $\dot{M}_0$ is then computed for each analysis window of *n* events (i.e., *n* = 200) and temporal length ($\Delta T$) as

$$\dot{M}_0 = \frac{\sum_{i=1}^{n} M_{0i}}{\Delta T} \tag{2}$$

### 2.4. b-Value and Completeness Magnitude $M_c$

The analysis of the cumulative frequency-magnitude distributions to estimate the *b-value* of the Gutenberg–Richter law [35].

$$\log N = a - b \times M_w, \tag{3}$$

where *N* is the cumulative number of earthquakes, *a* and *b* values are parameters describing the productivity and relative event size distribution, is carried out exploiting the entire-magnitude-range method [62] implemented in the software package ZMAP [63] and allows for the simultaneous estimate of the completeness magnitude $M_c$ and the parameters *a* and *b* (i.e., this latter obtained by the maximum likelihood approach [64]).

*2.5. Fractal Dimension*

The fractal dimension of earthquake hypocenters, $D_c$, is computed applying the correlation integral method [36,65]:

$$D_c = \lim_{r \to 0} \frac{\log C_r}{\log(r)} \tag{4}$$

where $r$ is the radius of a sphere of investigation and $C_r$ is the correlation integral:

$$C_r = \lim_{n \to \infty} \frac{1}{n^2} \sum_{i=1}^{n} \sum_{j=1}^{n} H\left(r - |x_i - x_j|\right) \tag{5}$$

with $n$ indicating the number of data in the analysis window (i.e., $n$ = 200 events), $x$ the hypocenter coordinates, and $H$ the Heaviside step function $H(x) = 0$ for $x \leq 0$ and $H(x) = 1$ for $x > 0$. In other words, $C_r$ is a function of the probability that two points will be separated by a distance less than $r$ [65].

The fractal dimension $D_c$ is therefore the slope of the best fit straight line of $C_r$ versus the distance $r$ in a bi-logarithmic diagram.

*2.6. Nearest-Neighbor Distance, η, Analysis*

The nearest-neighbor approach [66,67] is based on the computation of the generalized distance between pairs of earthquakes, $\eta$, from the analysis of the time-space distances between pairs of earthquakes. The parameter $\eta$ is derived computing the distances in time (i.e., rescaled time, $T$) and space (i.e., rescaled distance, $R$) between an event $i$ and its parent $j$ normalized by the magnitude of the parent event:

$$T_{ij} = t_{ij} 10^{-bm_i/2} \tag{6}$$

$$R_{ij} = \left(r_{ij}\right)^{D_c} 10^{-bm_i/2} \tag{7}$$

where $m$ is the magnitude ($M_w$), $b$ is the parameter of the Gutenberg–Richter law, $t$ is the earthquake intercurrence time, $r$ is the earthquake distance, and $D_c$ is the fractal dimension. The values of $b$ and $D_c$ are changed according to the estimates obtained for the considered window of events.

Finally, $\eta$ is defined as

$$\log \eta_{ij} = \log R_{ij} + \log T_{ij} \tag{8}$$

(see Zaliapin and Ben-Zion, 2016 [67] for further details).

*2.7. Shannon's Information Entropy*

The Shannon entropy, also known as information entropy (Shannon 1948) [38] provides a measure of the disorder level in a system. We compute the Shannon entropy using a regular 2-D grid ($m$ = 441 cells, each 1.1 km × 1.5 km).

We compute the Shannon entropy as

$$H = -\sum_{k=1}^{m} \frac{e_k}{E_R} \left[ \ln \frac{e_k}{E_R} \right] \tag{9}$$

where $e_k$ represents a fraction of the total seismic energy $E_R$ radiated within the $k$th cell [40]. The $e_k/E_R$ ratio is assumed to represent an empirical approximation of the probability of the seismic energy radiated in the $k$th cell, $P_k(E_R)$, with respect to the total radiated seismic energy, conditioned on the total energy radiated.

Equation (9) can therefore be rewritten as

$$H = -\sum_{k=1}^{m} P_k(E_R)[\ln P_k(E_R)] \tag{10}$$

Therefore, computing $H$ at a given temporal interval consists of summing up the $e_k/E_R$ ratio for the entire grid.

We computed the radiated seismic energy $E_R$ using the relation between local magnitude $M_L$ and seismic energy proposed for California by Kanamori et al., (1993) [68]:

$$E_R = 1.96 \times M_L + 2.05 \tag{11}$$

To allow comparison between different time intervals and to ensure unity total probability, $H$ is generally normalized to the equipartition entropy $H_E$, which corresponds to the case where $E_R$ is uniformly distributed in the cells (i.e., Equation (9) with $e_k = E/m$). The normalized information entropy $h = H/H_E$ ranges between 1 and 0, which correspond to the total disorder of the system and the extreme concentration of events, respectively.

The Shannon entropy concept provides hence a useful quantification of the system predictability; where $h = 0$ suggests the highest level of predictability and $h = 1$, on the contrary, suggests high disorder and low predictability.

## 3. Methods

In this section, we describe the RNN method that we developed to identify the preparatory phase of M4 earthquakes in The Geysers region and the aftershocks. For both goals, we used the same RNN algorithm, but we considered different sets of features as input and different targets. Each RNNs exploits five features, whereas only $\Delta t$ is used by both algorithms (Figure 3a). We call the configuration used to distinguish the preparatory phase RNN1, while we refer at the one used for the aftershocks phase as RNN2.

As said, we selected eight series of events including M4 earthquakes as case study. Hereinafter, each set of earthquakes is referred to as M4 series. The first five M4 series, in chronological order, were exploited as train sets and to optimize the method trough different validation techniques. Conversely, the test sets, consisting of the three more recent M4 series, was used downstream from all the other analyses to evaluate the methods performance and reliability.

### 3.1. Recurrent Neural Networks and ML Layers

Before going into details with the ML scheme, it is important to fix some ideas. An artificial neuron is a mathematical function that can be linear or not between input and output, which depends on coefficients whose optimization is done by a training. As example, a fully connected neuron gives as output a linear combination of all the inputs, adding also a bias. A neural network (NN) is an ensemble of neurons organized in various layers, each one with a certain number of neurons (also called nodes or units). The final model, in any case, will be a very complex mathematical function that can be very difficult to enroll and to understand. Despite that, the model will be almost completely data driven and so, if well trained, trustworthy.

Usually, NN layers are controlled by a set of hyperparameters that allows the model to fit the data. These hyperparameters are tuned by users, generally through a validation procedure. One of the most common hyperparameters is the learning rate. This latter is a small positive number that controls how much the model changes at each iteration, when a gradient descent technique is used to fit the data. Choosing a proper learning rate value is an important issue, because too small values can increase the computation time, conversely a too big one can hinder the model convergence.

In Figure 3b, we outline the functioning of an RNN. A key issue is that the RNNs see the data as connected sequential information. In our analysis, the data are time-connected, so the method looks at the information available at a given instant (i.e., used by RNN at $t_i$,

Figure 3b) and at the same time it keeps also that one from the previous step (i.e., that from RNN at $t_{i-1}$). This working strategy with memory of the past makes the output causal, and so particularly suitable for real-time applications. Despite that, the procedure proposed in this work is to be intended as off-line application; nevertheless, we will try to put it in real-time testing in future studies.

As we said, the NNs are organized in layers, so that they form the RNNs too. However, as can be seen in Figure 3c, the NN layers are usually different each other. In this study, we use two different kind of recurrent layers: fully connected RNN [69] and GRU [55]. Each RNN node (or unit, or neuron) has two inputs and two outputs. One of the RNN inputs comes from the previous layer, while the other input comes from the same unit at the previous instant. Similarly, one output goes to the same unit at next instant (hidden output), while the other output goes to the next layer. In a fully connected RNN unit, the hidden output is a linear combination of the two input, instead, the output towards the next layer is a linear combination of the inputs and of the hidden output. GRU is a more complex model than the previous one; it exploits logical gates to avoid the "vanishing gradient problem" [70].

Like we said, NN are formed by layers of multiple neurons. Since each neuron of a layer is connected with all the neurons of the next layer trough the output, this enormous number of relations can cause redundancy in the model, which can result into too deep models and overfitting. The dropout layers help to reduce the problem of overfitting [71]. These layers cut a certain number of connections between other two layers forcing the model to train in the simpler possible way. Each dropout layer is controlled by a dropout rate (DR), which represents the fraction of clipped connections.

Finally, our RNN includes an activation layer that simply transforms the input in an output through a mathematical operator (i.e., a sigmoid function, Figure 3c), and it is generally used to add non-linearity to the model.

Figure 3c provides a comprehensive view of the RNN scheme implemented in this work. The entire input dataset is a matrix $\left\{ N_f \times N_e \right\}$, where $N_f$ is the number of features and $N_e$ is the number of events for each M4 series. At each step, the model is fed with the features vector $\left\{ N_f \times 1 \right\}$ that contains the values computed at the current instant $t_i$. The first hidden layer is formed by $2 \cdot N_{node}$ GRUs. The outputs of this layer go through a dropout layer (characterized by a DR value). The third hidden layer is made by $N_{node}$ fully connected RNN nodes. After this latter layer, we put another dropout layer with the same DR as the one before. The fifth hidden layer is formed by a single fully connected NN node; this yields the final output through a sigmoid activation layer that allows the output to vary between 0 and 1. The output has s a single value for each instant, so it will be a vector $\{1 \times N_e\}$.

### 3.2. Step-By-Step Description of Data-Analysis by RNN

(1) Once the spatial-temporal characteristics (features) of seismicity have been extracted, we selected windows of data for the RNN analyses (i.e., 750 and 2000 earthquakes around the five M4 events for RNN1 and RNN2, respectively). In particular, for each M4 series in RNN1, we consider 499 events before the M4 event, the M4 itself, and 250 events after it. For RNN2, instead, we consider 1500 events before the M4, the M4 itself, and 499 after it. The different amount of data in the M4 series for RNN1 and RNN2 has been tuned for optimizing the training performance.

(2) Each M4 series has been standardized, which consist of, for each selected window, removing the mean and dividing for the standard deviation. This procedure is necessary since features span varying degrees of magnitude and units and these aspects can degrade the RNN performance [72] After the standardization, each feature is distributed as a standard normal distribution, N(0,1).

(3) In order to train the models, we assigned a label to each earthquake. Indeed, being RNN used here for sequential data classification, it is necessary train it with a simple binary (0/1) classification scheme. Therefore, in RNN1, the one aiming to identify the

preparatory phase, we assigned value 1 to those events preceding the M4s that have been interpreted based on expert opinion as belonging to the preparatory phase and label 0 to all the others (background and aftershocks). In RNN2, aiming to identify aftershocks, we assigned label 1 to aftershocks and label 0 to the others (background and foreshocks). In particular, in RNN1, for each M4 series, we selected a different number of events as belonging to the preparatory phase (i.e., ranging from 175 to 350 events) looking at the trend of parameters like *b-value*, $M_c$ and $Dc$ that we know likely related to the preparatory phase [27]. In RNN2, we decided to label all the 499 events following a M4 as aftershocks.

The rationale of our choice is that all features except two ($M_w$ and $\Delta t$) are computed on group of events (i.e., each window $n = 200$). Therefore, the features value at a given instant represents the collective behavior of a group of earthquakes, and it becomes not straightforward to single out, on this basis, if the last event of each window (i.e., the event at $t_i$) is a foreshock, or it belongs to the background seismicity (and similarly for aftershocks and background seismicity). Moreover, these considerations brought us also to avoid considering the location of these events (i.e., latitude, longitude and depth of the earthquakes are not used directly as features, but indirectly through $D_c$, $\eta$ and $h$ as collective characteristic of the seismicity), both for RNN1 and RNN2. Our aim during the training phase is to push RNN1 and RNN2 to find coherent correlation among features and to detect changes in the stationary character of the seismicity, which can correspond to the preparatory phase of a large earthquake and to aftershocks, respectively. It is important to note also that our strategy can be a two-edged sword, since the presence of misled background seismicity in both the preparatory and aftershock phases can lead to an underestimation of the performances falsely enhancing the number of missed events. In both cases, we thus make the RNNs work more challenging, and we test its capability to mine and detect coherent correlation among seismic features.

(4) The event catalogue is therefore transformed into two datasets (train and test) for RNN, in which lines contain the features and the labels (i.e., for line $i$, corresponding the at origin time of $i$th event, we have {$b\text{-}value_i$, $M_{ci}$, $D_{ci}$, $\Delta T_i$, $\dot{M}_{0i}$, $\Delta t_i$, $\eta_i$, $h_i$, $M_{wi}$, $label_{1i}$, $label_{2i}$}).

(5) We split the dataset in training and testing sets. The train set consists of the first five M4 series, while the testing one consists of the last three M4 series.

(6) We trained the model to find the best hyperparameters, which of course, could change between RNN1 and RNN2.

(7) The model has been validated on the train set separately (with the proper label) for RNN1 and RNN2 using a trial-and-error procedure to select the best features for the two models, and a leave-one-out (LOO) cross-validation to tune the hyperparameters. The LOO is performed leaving one M4 series per time and training the model on the other four. Hence, each model resulting from four M4 series is used to predict the target on the excluded M4 series. We decided to use AUC (area under the curve) as validation score because it is independent from the threshold selection. The mean of the five AUC values from the LOO validation is used to evaluate the hyperparameters configuration. We chose to explore three hyperparameters, which are $N_{node}$ (in the range 3–20), DR (between 0 and 0.5), and the learning rate LR (between $1 \times 10^{-5}$ and $1 \times 10^{-3}$) with which the model is fitted.

(8) A similar leave-one-out (LOO) approach has been carried out also to perform a feature importance analysis. In this case, considering the five M4 series, we proceeded at removing one by one features and checking the performance with respect to the case with all features. On one hand, our tuning led to select for RNN1 the features *b-value*, $M_c$, $D_c$, $\Delta T$ and $\Delta t$. On the other hand, we selected for RNN2 to features $\dot{M}_0$, $\Delta t$, $\eta$, $h$ and $M_w$.

(9) The performance of the RNN1 and RNN2 models has been assessed using the testing dataset of three M4 series.

## 4. Results

### 4.1. Observing Seismicity Occurrence from Features Perspective

As discussed, we took advantage of the information on event date and time of occurrence, location, depth and local magnitude extracted from The Geysers' seismic catalogue to observe the temporal and spatial evolution of seismicity. To study the frequency magnitude events distribution in terms of the Gutenberg–Richter relationship (i.e., *b-value* and $M_c$), we converted the local magnitude, $M_L$, in moment magnitude scale, $M_w$. The $M_L$ scale has also been used to derive estimates of the radiated energy, $E_R$, which in turn has been utilized to compute the Shannon's information entropy, *h*. These pieces of information were transformed in data time features: *b-value*, $M_c$, Shannon's information entropy (*h*), fractal dimension ($D_c$), generalized distance between pairs of earthquakes ($\eta$), Moment rate ($\dot{M}_0$), inter-event time ($\Delta t$), total duration of event groups ($\Delta T$), and $M_w$.

The first M4 event recorded in the area occurred on May 2006. The temporal evolution of features associated to earthquakes preceding this event shows, approximately the day before, peculiar trends with respect to the preceding background seismicity, which for the sake of simplicity have been highlighted (Figure 4, red values). For most of features, these trends correspond to a more-or-less pronounced decrease (e.g., *b-value*, $M_c$, $D_c$, $\Delta T$, *h*, $\eta$, $\Delta t$), while $\dot{M}_0$ it increases.

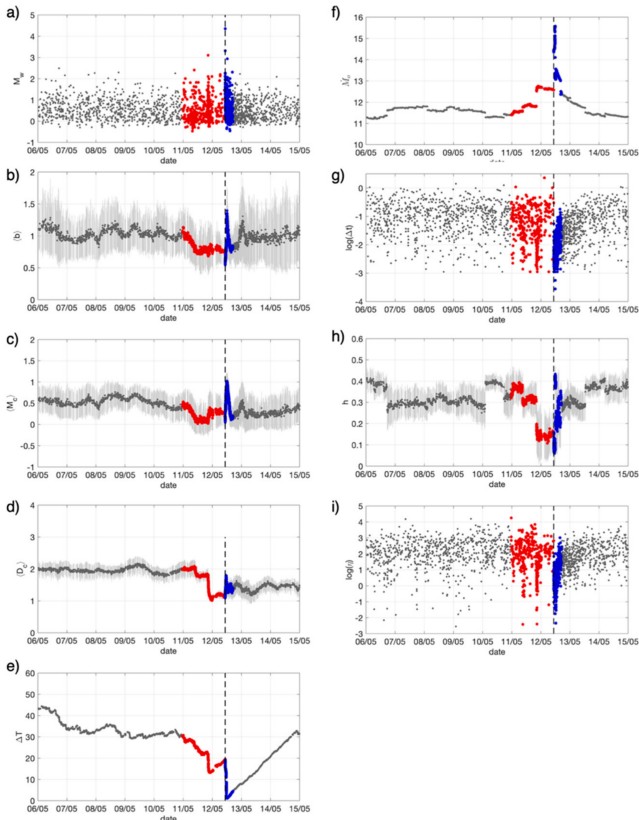

**Figure 4.** (**a**) Distribution of events before and after the first M4 earthquake (occurred the 12th May 2006, vertical dashed line). Background seismicity (grey dots), foreshocks (red), main-event and aftershocks (blue). All the other plots are the same as (**a**), but for: (**b**) average *b-values* and $\pm 1$ SD bars (grey); (**c**) average completeness magnitude, $M_c$, and $\pm 1$ SD bars; (**d**) average fractal dimension, $D_c$, and $\pm 1$ SD bars; (**e**) total duration, $\Delta T$, of the events group; (**f**) Moment rate, $\dot{M}_0$; (**g**) inter-event time, $\Delta t$; (**h**) Shannon entropy, *h*; (**i**) Logarithm (base 10) of the generalized distance between pairs of earthquakes, $\eta$. The uncertainty associated to *b-value*, $M_c$, $D_c$ and Shannon entropy *h* is computed by means of a bootstrap approach [73], whereas for each dataset 1000 realizations of random sampling with replacement were performed.

In addition to the events preceding the M4, we found peculiar also the features trend after it (Figure 4, blue values). The features show, indeed, considerable decrease or increase also for aftershocks.

The standardized features for the first five M4 earthquakes (Figure 5), occurred between 2006 and 2012, show coherent trends (i.e., *b-value*, $M_c$, $D_c$, $\Delta T$ and $\Delta t$ for the phase preceding the mainshocks, while $\dot{M}_0$, $\Delta t$, $\eta$, $h$ for the period following it).

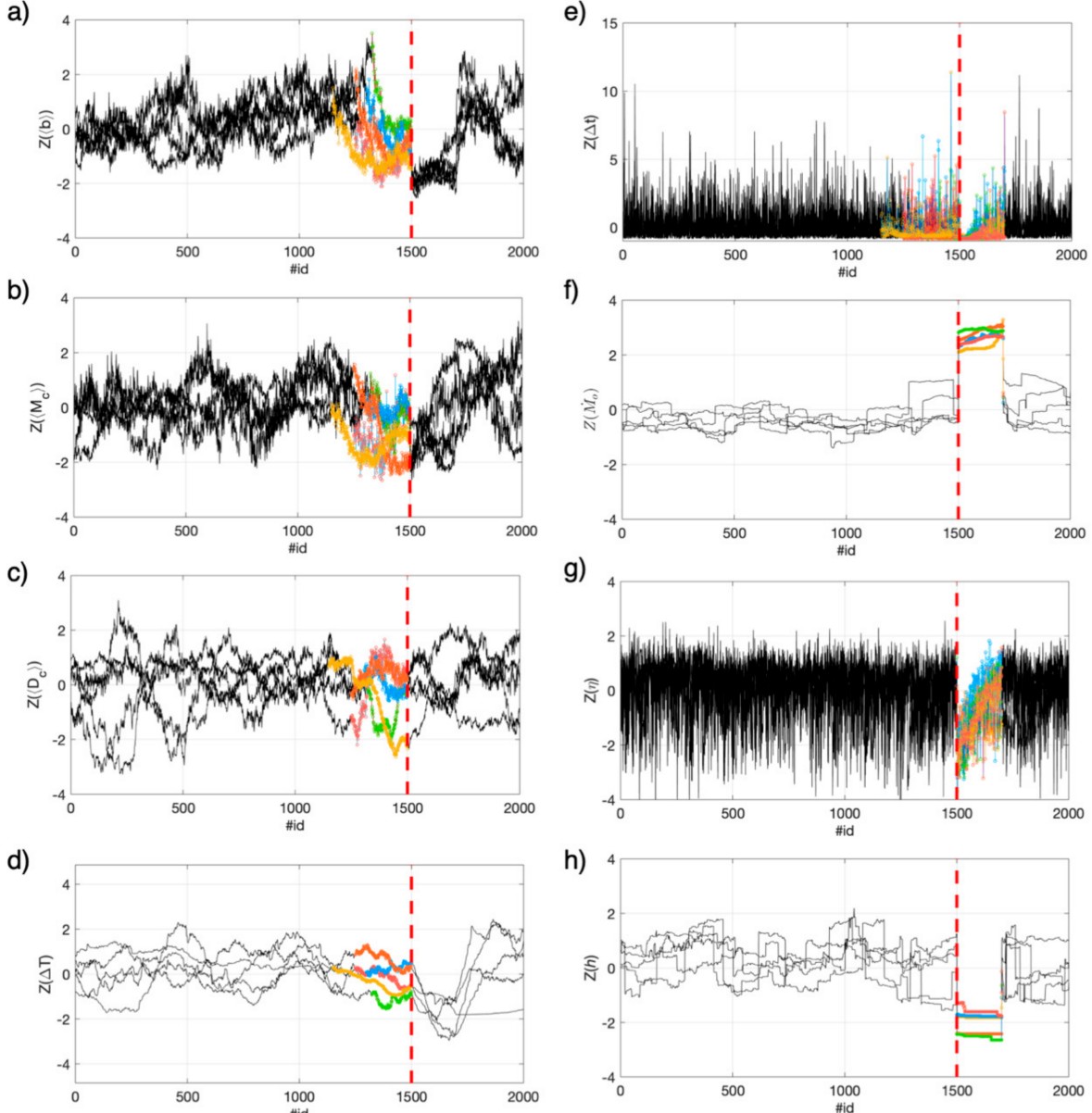

**Figure 5.** Features belonging to the training M4 series after standardization. Background seismicity (black), foreshocks and aftershocks (colored per event). Time of the M4s occurrence (vertical, dashed red line). Subplots show different features according to Figure 4.

Following these observations, we decided to explore the capability of RNN in capturing the long-term trends in data and the, likely non-linear, correlation among features.

With respect to the trend associated to background seismicity, both what we assume being the preparatory phase and the aftershocks occurrence represent transients. While the automatic discrimination of aftershocks is for monitoring systems interesting in itself, this is especially true for us. As a matter of fact, the occurrence of transients associated to aftershocks might be confusing for the identification of transients associated to the

preparatory phase of large earthquakes from background seismicity. For this reason, as discussed in Section 3, we decided to train two RNNs: one dedicated to the preparatory phase (RNN1), and another aiming to identify the aftershocks (RNN2). In the following, we describe first how RNNs have been trained and tested. Finally, we discuss how they could be combined for induced seismicity monitoring purpose at The Geysers.

### 4.2. RNNs Tuning Trough Cross-Validation on a Training Dataset

Figure 6a shows the events labelled zero (i.e., belonging to stationary seismicity and aftershocks sequence, black dots) and one (i.e., belonging to the preparatory phase, green dots). Similarly, in Figure 6b, we show for the first five M4 series the events labelled zero (i.e., background and foreshocks, black dots) and one (i.e., aftershocks, green dots)

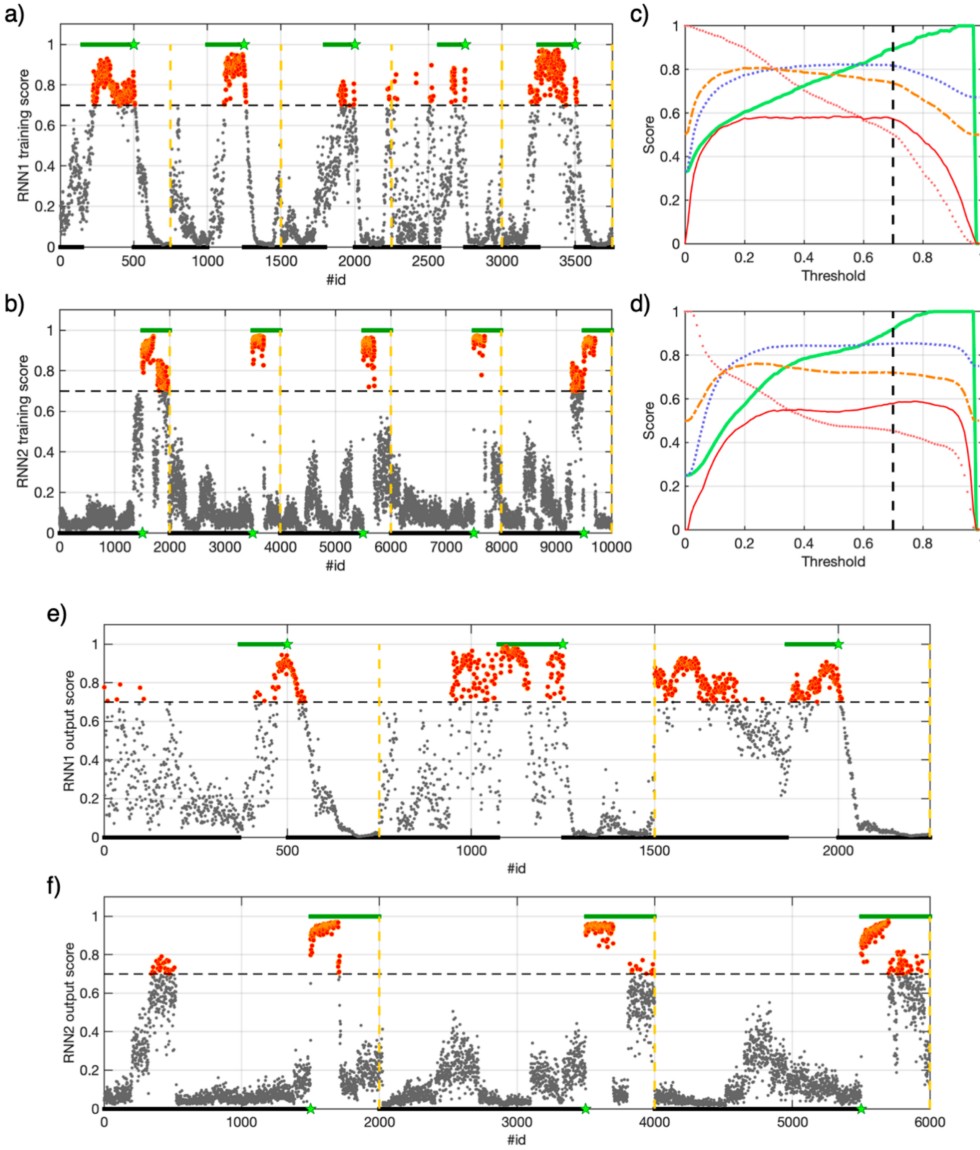

**Figure 6.** (**a**) RNN1 training. Labels for background and aftershocks (black), while for preparatory events (green), M4 events (green stars), end of the M4 series (vertical, dashed yellow line), selected threshold (horizontal, dashed black line), RNN1 output below the threshold (grey), and above it (orange). (**b**) The same as (**a**), but for RNN2 and with label for background and foreshocks (black), while for aftershocks (green). (**c**) Performance criteria versus threshold: recall (red dotted line), precision (green thick line), accuracy (blue square dotted line), balanced accuracy (dashed orange line), and MCC (red thin line) for RNN1. Selected threshold (vertical, dashed black line). (**d**) The same as (**c**), but for RNN2. (**e**) The same (**a**) but for the testing dataset and RNN1 (three M4 testing series). (**f**) The same as (**b**) but for the testing dataset and RNN2.

To tune the RNNs parameters, we have carried out a leave-one-out (LOO) analysis using the five-training series (i.e., each run, four M4 series are in and one is out). As discussed in Section 3.2, we avoided to select a priori a threshold value, and we decided to study the variation of the AUC parameter (i.e., the area under the receiver operating characteristic, ROC, curve) as score of performance. The average of the five AUC values from the LOO has been used to select the best RNN1 and RNN2 models.

We also performed a features importance analysis following an approach similar to LOO for assessing which features were optimizing the capability of RNN1 and RNN2 in identifying their targets. The results indicate that for RNN1 the best features are *b-value*, $M_c$, $D_c$, $\Delta T$ and $\Delta t$; while for RNN2 it is better to use $\dot{M}_0$, $\Delta t$, $\eta$, $h$ and $M_w$.

### 4.3. Training RNN1 and RNN2

In Figure 6c,d, we show the trends of different performance criteria with the respect of the threshold used in the binary classification for RNN1 and RNN2, respectively. The ensemble of these criteria gives us a comprehensive statement on the model performances. To better understand why, let us introduce these criteria using the binary classification (positive/negative event): "Recall" represents how many positive events are well-classified; "Precision", instead, is the rate of true positive on all the events classified as positive; "Accuracy" represents the rate of true predictions on the total of the predictions; "Balanced Accuracy" is the mean between the recall and the rate of negative events well-classified (also called selectivity, it is similar to Accuracy, but it does not suffer the unbalanced dataset); "MCC", Matthew correlation coefficient [74,75] is related to the general quality of the classification (we will dedicate more space to this specific parameter in the following).

Selecting the threshold equal to 0.7 for both RNN models, we think to have found a best compromise in terms of different performance criteria (Figure 6c,d). Therefore, we used this threshold for both the RNN1 and RNN2 best models applied again to the five M4 training series (Figure 6a,b,e,f).

Concerning the preparatory phase, we observe that a significant portion of the events with label 1 are correctly identified (i.e., for series "1", "2", and "3" the score overcome 0.7 in 74%, 53%, and 68% of the cases, respectively). Only for the third and fourth time series the number events considered as belonging to a preparatory phase is reduced, but still the 14% and 18%, respectively (Figure 6a).

Similarly, also the RNN2 performance is convincing, since a high number of after-shocks has been correctly identified for all data series (Figure 6b). In this case, success rate varies between the 40% and the 66%. To assess the quality of our binary classification, we also exploited the Matthews correlation coefficient (MCC), which taking into account the rate of negatives and positives provides a measure of the correlation coefficient between the observed and predicted classifications [74,75]. MCC assumes value ranging between $-1$ and $+1$; whereas MCC equal to $+1$ represents a perfect prediction, equal to 0 indicates no better than a random prediction, and finally equal to $-1$ indicates total disagreement between prediction and observation.

For RNN1, we obtain MCC equal to 0.74, 0.60, 0.29, 0.26, and 0.66 for the testing M4 series number from 1 to 5, respectively. For RNN2, we obtain MCC equal to 0.77, 0.57, 0.57, 0.57, and 0.40 for M4 series number 1 to 5, respectively. These results indicate that RNN1 and RNN2 show, with respect to our labels assignment, a good prediction capability.

In addition to the MCC scores, interestingly, we note that both before and after the mainshock a continuous set of events are classified as preparatory and aftershock phases, but for a total number smaller than the one considered by us. This means that the selected seismic features lead RNN1 and RNN2 be able to autonomously declare the preparatory phase be initiated and the aftershock sequence concluded, respectively.

### 4.4. Testing RNN1 and RNN2

Intrigued by the training outcomes, we tested our RNNs on the remaining three M4 series, which have occurred between 2013 and 2014.

RNN1 results capable of identifying all three preparatory phases (success rate is 28%, 58%, and 77% for events with label 1, respectively, Figure 6e). It is worth noting that RNN1 assigns score larger than the threshold also to events before what we defined from visual inspection as preparatory phase, but we will come back to this point later.

RNN2 provides good classification results too (Figure 6f); indeed, in this case the success rate in identifying aftershocks is 42% for the first two M4 testing series and 50% for the third one.

For RNN1, we obtain MCC equal to 0.251, 0.42, and 0.346 for the testing M4 series number 1, 2, and 3, respectively. For RNN2, we obtain MCC equal to 0.534, 0.597, and 0.653 for M4 series number 1, 2, and 3, respectively.

These results indicate that RNN1 and RNN2 provide good, not random predictions. We consider worth to remember, that the MCC scores is computed with respect to our labels assignment, which we kept large on purpose and could include together with earthquakes of the preparatory phase and aftershocks also events belonging to the background seismicity (see Section 3.2 issue #3).

Similarly to the training case, we observe that RNN1 and RNN2 lead the preparatory phase and aftershock sequences start after and conclude before, respectively, with respect our labels assignment. However, once a non-stationary phenomenon is detected, the classification does not jump from 0 to 1 randomly, but RNNs hold it stably; indicating that the RNNs are able to autonomously discriminate the events belonging to the preparatory phase and aftershocks sequence from those of the stationary, background seismicity.

### 4.5. Conceptualization of an Alert Algorithm

The goal of our study is to show that the identification of the preparatory phase of large earthquakes is becoming an achievable task when the temporal and spatial evolution of seismicity is monitored by dense local networks. Although we are only moving the first steps towards the possibility of implementing an operational algorithm for earthquake forecasting, here we conceptualize an application of our RNN models for detecting when the seismicity is evolving along the preparatory phase of large magnitude event at The Geysers.

Our scheme considers the RNN scores as probability values (i.e., in RNN1 the score represents the probability for the event of belonging to the preparatory phase, while in RNN2 it means the probability of being an aftershock).

We thus combine the two RNN probabilities in an alert probability ($P_{ALERT}$) as follows:

$$P_{ALERT i} = P_{RNN1 i} \times \left(1 - P_{RNN2(i-9:i)}\right) \tag{12}$$

where $P_{RNN1 i}$ is the probability associate to event $i$ and $P_{RNN2(i-9:i)}$ is the average of probabilities associated to $i$ and the previous nine events. The need to consider the average of ten $P_{RNN2}$ is related to the aim of stabilizing the $P_{ALERT}$ for a given event. Indeed, as can be seen in Figure 6b or Figure 6f, even if for background seismicity and foreshocks RNN2's outputs are very small (i.e., between 0 and 0.4), every change in $P_{RNN2}$ would lead to oscillation in $P_{ALERT}$. It is worth to remember that Equation (12) represents just a simple example of possible criteria for combining the RNNs' outputs. The proposed alert system outline is just an example which aims to show a potential future application of a similar research RNN products and to stimulate the debate about strategies for identifying the preparatory of induced and natural large earthquakes.

The temporal evolution of $P_{ALERT}$ for the three M4 testing series shows interesting results (Figure 7). For the first M4 series, before the 14th March, the selected probability threshold (i.e., $P_{thr} = 0.7$) is overcome by $P_{ALERT}$ only rarely. Then, interestingly, $P_{ALERT}$ overcomes stably the selected probability threshold around four hours before the mainshock (Figure 7a). For the second and third M4 series, $P_{ALERT}$ overcome rather stably $P_{thr}$ two and four days before the mainshocks, respectively. Moreover, in both cases, there is a drop

in $P_{ALERT}$ which is followed by a rapid increase above $P_{thr}$ few hours before the mainshock (Figure 7b,c).

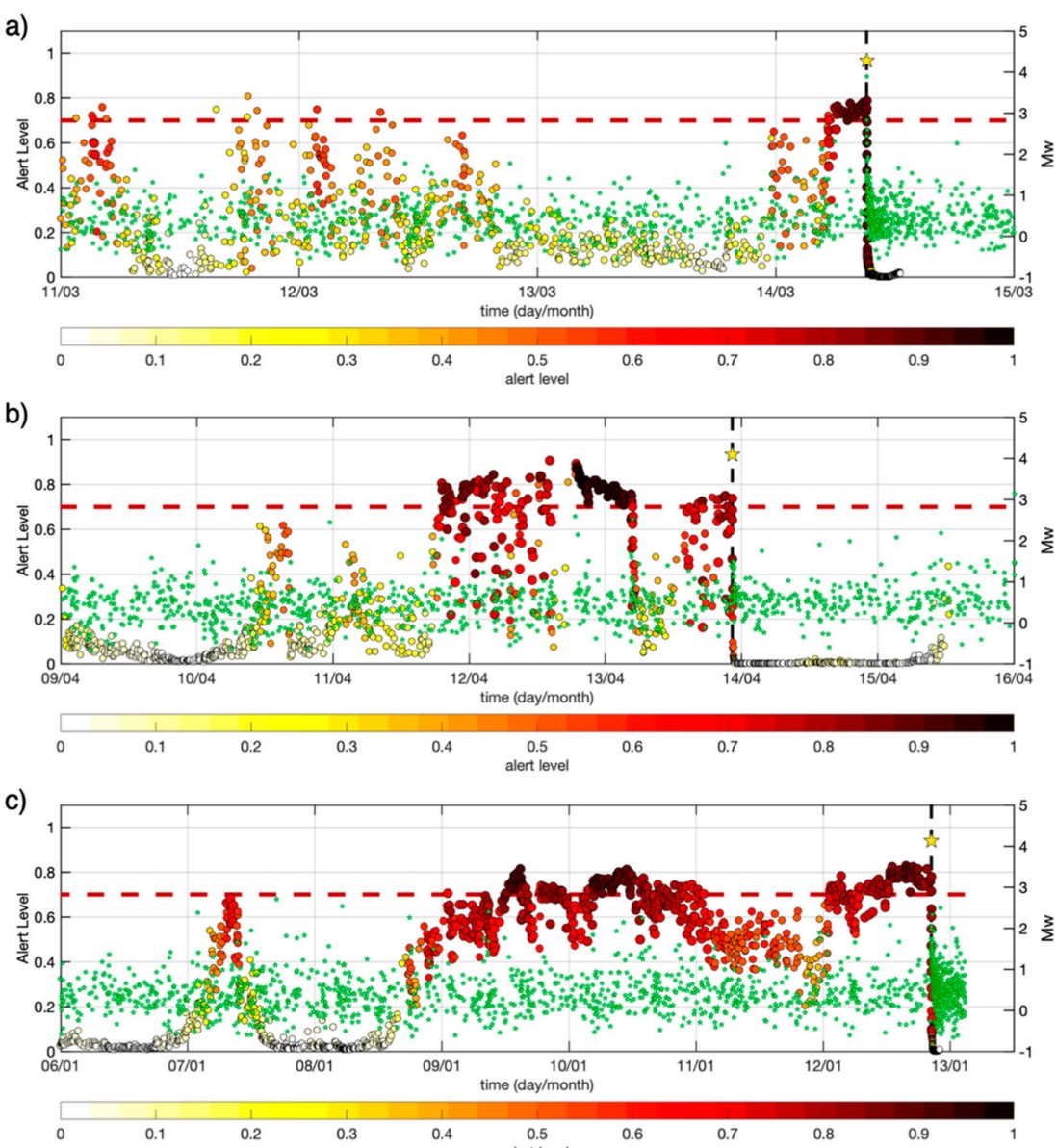

**Figure 7.** Application of RNN1 and RNN2 combined into a prototype of alert system to the three M4 testing series. (**a**) Distribution of event magnitude in time (green stars and right-side *y*-axis) for the M4 that occurred on the 14th March 2013 (yellow star). The threshold alert level (dashed red line) and alert probability ($P_{ALERT}$) refer to the left-side *y*-axis. $P_{ALERT}$ are dots colored and with size varying according to their values (i.e., from white, no alert, to dark red for events above the threshold alert level). (**b**) The same as (**a**), but for the M4 that occurred on the 14th April 2013. (**c**) The same as (**a**), but for the M4 that occurred on the 13th January 2014.

## 5. Discussion

Given the very high rate of seismicity (>10,000 earthquakes/year) and the high-quality seismic network, The Geysers represents a perfect natural laboratory for studying the rupture processes and nucleation of induced earthquakes. The interaction of industrial activities associated to many injection wells with the tectonic stress suggests a complex stress-field, a puzzle difficult to resolve with standard approaches. Our results indicate that the spatial and temporal evolution of seismicity can be unveiled by benefiting of only information obtainable from seismic catalogue and using machine learning.

The seismicity in the area occurs mostly at shallow depth in relation to the ~70 water injection wells and predominantly related to thermal stress [76]. However, the largest events (M4) occur confined in the deepest portion of the reservoir, have relatively high stress drop values and result spatially distributed in agreement with the regional fault trend [61], characterized by right-lateral strike-slip faults. Their occurrence is therefore interpreted as due to the reactivation of deep structures linked to regional tectonics [77]. It is out of the scope of this work to investigate the physical mechanism driving the generation o M4 events and settle the controversy between earthquake foreshock theories.

Our application at The Geysers suggests, however, that the M4 earthquakes are preceded by preparatory process of the mainshocks compatible with the loading view [9–11,26].

Our results show that the preparatory phase for the three testing M4 earthquakes lasted from few hours to few days, in agreement with the short-time preparation process (~1 day) observed for a similar magnitude natural earthquake ($M_w$ 4.4) occurred recently in Italy [29]. Future work will explore the application of RNN to the real-time identification of the preparatory phase at The Geysers and other areas where induced seismicity occurs.

Furthermore, we will test whether seismic derived features as those used in this study can allow to identify the preparatory phase of large magnitude natural earthquakes too. Our approach can potentially set marked advances in the earthquake predictability research.

**Author Contributions:** M.P. and A.G.I. conceived the analysis method, performed the data analysis, created the figures, and wrote the paper. All authors have read and agreed to the published version of the manuscript.

**Funding:** This work was carried out in the framework of the project "Detection and tracking of crustal fluid by multi-parametric methodologies and technologies" [code 20174X3P29], funded by PRIN: PROGETTI DI RICERCA DI RILEVANTE INTERESSE NAZIONALE—Bando 2017.

**Institutional Review Board Statement:** Not applicable.

**Informed Consent Statement:** Not applicable.

**Data Availability Statement:** Data available in a publicly accessible repository. The data presented in this study are openly available in 'The Northern California Earthquake Data Center (NCEDC)' at http://service.ncedc.org/fdsnws/dataselect/1/.

**Acknowledgments:** The authors would also like to thank the Editor N. Jiang and two anonymous reviewers for comments and suggestions.

**Conflicts of Interest:** The authors declare no conflict of interest.

**Resources:** We used the following ML resources: **KERAS** (Chollet, F., Keras. https://github.com/fchollet/keras, 2015); **TENSORFLOW** (M. Abadi, A. Agarwal, P. Barham, E. Brevdo, Z. Chen, C. Citro, G.S. Corrado, A. Davis, J. Dean, M. Devin, S. Ghemawat, I. Goodfellow, A. Harp, G. Irving, M. Isard, R. Jozefowicz, Y. Jia, L. Kaiser, M. Kudlur, J. Levenberg, D. Mané, M. Schuster, R. Monga, S. Moore, D. Murray, C. Olah, J. Shlens, B. Steiner, I. Sutskever, K. Talwar, P. Tucker, V. Vanhoucke, V. Vasudevan, F. Viégas, O. Vinyals, P. Warden, M. Wattenberg, M. Wicke, Y. Yu, and X. Zheng. TensorFlow: Large-scale machine learning on heterogeneous systems, 2015. Software available from tensorflow.org.); **SCIKIT-LEARN** (Pedregosa, F. et al., 2011. Scikit-learn: Machine learning in Python. Journal of machine learning research, 12(Oct), pp.2825–2830).

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
