# Peer review of "Forecasting the Preparatory Phase of Induced Earthquakes by Recurrent Neural Network"

_forecasting, doi:10.3390/forecast3010002_

Round 1

Reviewer 1 Report

            This paper uses a recurrent neural network to investigate foreshock and aftershock activity of M~4 induced earthquakes in the Geysers geothermal field.  It is an intriguing study, but my general concern is that there is not enough detail in the paper to really understand the method and confirm the results, especially for those readers who may not be as familiar with machine learning/recurrent neural networks.  This makes parts of the paper more difficult to follow and interpret.  It would be helpful to spend a little bit more space explaining some terms that a broader audience might not be familiar with.  Some specific areas where this could be particularly helpful are listed below, with more minor comments.

Figure 1: The labels for (b) and (c) in this figure seem to be switched.

l. 81: It would be helpful if you could just list out the 10 features here.

l. 132: Why is the specific value of n=200 used?  Did you try other values?  What is the sensitivity of the results to this parameter?

ll. 158-159: You should also cite the original paper describing the entire-magnitude-range method:

Woessner, J., and S. Wiemer (2005), Assessing the quality of earthquake catalogues: Estimating the magnitude of completeness and its uncertainty, Bull. Seismol. Soc. Am., 95, 684-698, doi: 10.1785/0120040007.

ll. 222-223: What exactly do you mean by “standardized”?

ll. 224-228: How exactly do you distinguish between whether an event is a background event vs. an event that is part of the preparatory phase?  Or between a background event and an aftershock, for that matter.  For example, just eyeballing the change of density of events after the mainshock in Fig. 2a, it looks like the aftershocks might actually continue beyond the blue time period.

ll. 242-259: I think more details would be useful in this section, such that someone who isn’t well versed in RNNs can understand what exactly is going on in the scheme.  For example, what is a dropout layer?  Why is it used?  What is the learning rate?

ll. 305-307: I have a really hard time seeing many coherent trends in the periods preceding the mainshocks in Fig. 3.  It just looks quite noisy.  Can you be more specific?  Which trends do you think are the most convincingly coherent?  And I still do not see any convincing differences between background events and foreshocks.

ll. 345-347: Please include brief descriptions of the different performance criteria (recall, precision, accuracy, balanced accuracy).

ll. 359-365: I am having some difficulty understanding and interpreting Fig. 4, and this is probably another section that could use more detail.  Taking Fig. 4a, for example, if I understand correctly, everything above the 0.7 threshold is identified by RNN1 as a foreshock, but what exactly is the difference between the green and the orange?  Does the green indicate events that were actually foreshocks?  It would help to describe how “success rate” is defined, perhaps I’m just misunderstanding it.  For example, how is the success rate for RNN2 (between 40-66%) any better than random chance?

ll. 387-388: Why the previous 9 events?

Author Response

REVIEWER #1

We thank the Reviewer for the comments he/she provided, which have helped us to improve the quality of the manuscript.

We have included all suggestions into the revised manuscript.

Reviewer’s General comment: This paper uses a recurrent neural network to investigate foreshock and aftershock activity of M~4 induced earthquakes in the Geysers geothermal field.  It is an intriguing study, but my general concern is that there is not enough detail in the paper to really understand the method and confirm the results, especially for those readers who may not be as familiar with machine learning/recurrent neural networks.  This makes parts of the paper more difficult to follow and interpret.  It would be helpful to spend a little bit more space explaining some terms that a broader audience might not be familiar with.  Some specific areas where this could be particularly helpful are listed below, with more minor comments.

Answer: We thank the Reviewer for the suggestion. We included two new figures (new Fig1 and new Fig3) to better outline the general model of the processes anticipating and following a large earthquake that we try to catch by ML and the RNN method used in this study. The description of the method is presented in section 3.

Question Figure 1: The labels for (b) and (c) in this figure seem to be switched.

Answer: Corrected

Question l. 81: It would be helpful if you could just list out the 10 features here.

Answer: As suggested, we included the list of features into the Introduction

Question l. 132: Why is the specific value of n=200 used?  Did you try other values?  What is the sensitivity of the results to this parameter?

Answer: We performed preliminary testing for selecting a unique number of events for a given window (n) that could be adequate for all features, being aware of competing needs. Indeed, a rapid detection of transient signals within a background stationary trends would require computing the features with small samples of data, while the stability of some features is strongly influenced by the number of data (i.e., as for instance discussed by Marzocchi et al., 2020, included in the references, for the b-values). Our choice of using n=200 is thus a compromise. For instance, Dresen et al., 2020 [27], used n equal to 150.

Question ll. 158-159: You should also cite the original paper describing the entire-magnitude-range method: Woessner, J., and S. Wiemer (2005), Assessing the quality of earthquake catalogues: Estimating the magnitude of completeness and its uncertainty, Bull. Seismol. Soc. Am., 95, 684-698, doi:10.1785/0120040007.

Answer: Thank you for having highlighted the reference. We added it.

Question ll. 222-223: What exactly do you mean by “standardized”?

Answer: “standardized” refers to the operation applied to each selected windows. It consist of removing the mean and dividing for the standard deviation. This procedure is necessary in ML since features span varying degrees of magnitude and units and these aspects can degrade the RNN performance [please, see also reference 72]. After the standardization, each feature is distributed as a standard normal distribution, N(0,1).

Question ll. 224-228: How exactly do you distinguish between whether an event is a background event vs. an event that is part of the preparatory phase?  Or between a background event and an aftershock, for that matter.  For example, just eyeballing the change of density of events after the mainshock in Fig. 2a, it looks like the aftershocks might actually continue beyond the blue time period.

Answer: This is an important issue for the work, and we thank the Reviewer for having highlighted. We included new text to explain the adopted approach in section 3.2.

We report in the following the text for the Reviewer’s convenience:

(3) In order to train the models, we assigned a label to each earthquake. Indeed, being RNN used here for sequential data classification, it is necessary train it with a simple binary (0/1) classification scheme. Therefore, in RNN1, the one aiming to identify the preparatory phase, we assigned value 1 to those events preceding the M4s that have been interpreted based on expert opinion as belonging to the preparatory phase and label 0 to all the others (background and aftershocks). In RNN2, aiming to identify aftershocks, we assigned label 1 to aftershocks and label 0 to the others (background and foreshocks). In particular, in RNN1, for each M4 series, we selected a different number of events as belonging to the preparatory phase (i.e., ranging from 175 to 350 events) looking at the trend of parameters like b-value, Mc and Dc that we know likely related to the preparatory phase [27]. In RNN2, we decided to label all the 499 events following a M4 as aftershocks.

The rationale of our choice is that all features except two (Mw and Dt) are computed on group of events (i.e., each window n = 200). Therefore, the features value at a given instant represents the collective behavior of a group of earthquakes, and it becomes not straightforward to single out, on this basis, if the last event of each window (i.e., the event at ti) is a foreshock, or it belongs to the background seismicity (and similarly for aftershocks and background seismicity). Moreover, these considerations brought us also to avoid considering the location of these events (i.e., latitude, longitude and depth of the earthquakes are not used directly as features, but indirectly through Dc, h and h as collective characteristic of the seismicity), both for RNN1 and RNN2. Our aim during the training phase is to push RNN1 and RNN2 to find coherent correlation among features and to detect changes in the stationary character of the seismicity, which can correspond to the preparatory phase of a large earthquake and to aftershocks, respectively. It is important to note also that our strategy can be a two-edged sword, since the presence of misled background seismicity in both the preparatory and aftershock phases can lead to an underestimation of the performances falsely enhancing the number of missed events. In both cases, we thus make the RNNs work more challenging, and we test its capability to mine and detect coherent correlation among seismic features.

Question ll. 242-259: I think more details would be useful in this section, such that someone who isn’t well versed in RNNs can understand what exactly is going on in the scheme.  For example, what is a dropout layer?  Why is it used?  What is the learning rate?

Answer: As requested, we included the new figure (3) and section 3 for better explaining the RNN and GRU methods.

Question ll. 305-307: I have a really hard time seeing many coherent trends in the periods preceding the mainshocks in Fig. 3.  It just looks quite noisy.  Can you be more specific?  Which trends do you think are the most convincingly coherent?  And I still do not see any convincing differences between background events and foreshocks.

Answer: following the Reviewer’s suggestion, we try to simplify the old figure (3), now figure (5), highlighting with different colors for each feature only the events that are used in RNN1 or RNN2. For instance, for b-value, Dc, Mc and Dt we colored only events preceding the M4 earthquake that we assume belongs to a preparatory phase (used by RNN1); while for h, Shannon’s entropy and moment rate, we colored only the aftershocks, because these features are used only by RNN2. The only features exploited by both RNNs is Dt.

Question ll. 345-347: Please include brief descriptions of the different performance criteria (recall, precision, accuracy, balanced accuracy).

Answer: Done.

Question ll. 359-365: I am having some difficulty understanding and interpreting Fig. 4, and this is probably another section that could use more detail.  Taking Fig. 4a, for example, if I understand correctly, everything above the 0.7 threshold is identified by RNN1 as a foreshock, but what exactly is the difference between the green and the orange?  Does the green indicate events that were actually foreshocks?  It would help to describe how “success rate” is defined, perhaps I’m just misunderstanding it.  For example, how is the success rate for RNN2 (between 40-66%) any better than random chance?

Answer: In the old Figure 4a, now figures 6a, we show the temporal evolution of the labels assigned by us (0 = black dots for background, 1 = green dots for preparatory phase) with respect to the RNNs output (gray for events with output below the 0.7 threshold and classified as belonging to the background seismicity; orange for output equal or above 0.7, which are classified as belonging to the preparatory phase).

For discussing the success rate and to show that the classification is better than random chance, we used the Matthews correlation coefficient (MCC), which taking into account the rate of negatives and positives provides a measure of the correlation coefficient between the observed and predicted classifications [see references 74, 75]. MCC assumes value ranging between -1 and +1; whereas MCC equal to +1 represents a perfect prediction, equal to 0 indicates no better than a random prediction, and finally equal to −1 indicates total disagreement between prediction and observation. 

For RNN1 during the training phase, we obtain MCC equal to 0.74, 0.60, 0.29, 0.26, and 0.66 for the testing M4 series number from 1 to 5, respectively. For RNN2, we obtain MCC equal to 0.77, 0.57, 0.57, 0.57 and 0.40 for M4 series number 1 to 5, respectively.

During the testing phase, for RNN1, we obtain MCC equal to 0.251, 0.42 and 0.346 for the testing M4 series number 1, 2, and 3, respectively. For RNN2, we obtain MCC equal to 0.534, 0.597 and 0.653 for M4 series number 1, 2, and 3, respectively.

These results indicate that RNN1 and RNN2 provide good, not random predictions. We consider worth to remember, that the MCC scores is computed with respect to labels we assigned, we kept these large on purpose and could include together with earthquakes of the preparatory phase and aftershocks also events belonging to the background seismicity (see section 3.2 issue #3).

We observe that RNN1 and RNN2 lead the preparatory phase and aftershock sequences start after and conclude before, respectively, with respect our labels assignment (Figure 6). However, once a non-stationary phenomenon is detected, the classification does not jump from 0 to 1 randomly, but RNNs hold it rather stably; indicating that the RNNs are able to autonomously discriminate the events belonging to the preparatory phase and aftershock sequence from those of the stationary, background seismicity.

Question ll. 387-388: Why the previous 9 events?

Answer: Looking at Figure 6b or 6f, it can be seen that for the background seismicity the RNN2’s outputs are generally very small but varying between 0 and 0.4. Thus, in order to stabilize the probability of an alert in Eq.(12), we considered 10 events for computing the average of PRNN2 and not the use of a single RNN2’s output. The value of 10 has been found by trial and error, but it is just an example for discussing the possible future application of ML in assisting decision systems, which we imagine will include cost-benefit analysis criteria and other principles of risk analysis.

Reviewer 2 Report

In this manuscript, the authors have proposed to detect earthquake preparation phase using RNN. They have computed seismic features using the temporal sequence of past seismicity and detecting the class of subsequent earthquake (foreshock / background). The idea is interesting. However, it needs some revision in presentation and analysis before acceptance. 

Comments from writing perspective:

1- Being a reader, I am unable to understand the meaning of Line 110-115. Please elaborate. In particular, line 114-115 feels a bit distracting. If you are not including a certain precursor, why feel the need to put it here. Just go with what you are already doing. 

2- Line 135-136: I feel these lines need more clear explanation.

3- Equation 2: Does this equation mean that sum of moment magnitude of 200 events divided by delta T? If Yes, then perhaps the notation needs changing from Mo to Mo i.

4- Line 218-219: What makes you choose 750 and 2000 around five M4 events?

5- Line 226-227: Why switch labelling technique and assign label 1 to aftershocks and 0 to others?

6- I feel that there is lack of flow in the manuscript. However, I can be wrong. May I suggest that a clear and separate section or subsection may be created to explain dataset formation. Then a new section or subsection may be made to discuss on RNN application. There should be clear segregation between explanation of dataset and RNN.

Since your state of the art is not RNN. Your state of the art is this idea and formation of dataset in this the said manner, thus make a clear section of it.

Comments from analyses perspective:

1- How are you able to identify foreshocks, aftershocks and and background seismicity? Being a student, I think that there are multiple methods for catalog deculstering and their results may vary. Since you do not refer to the method you have used for deculstering or identify foreshocks, aftershocks and background events. It should be referred at least.

2- Line 251-253: It says that Test set is used for Validation. Please ensure that Testing takes place independently of validation. Use different part of dataset for validation. Testing must be done on completely different dataset, which was otherwise not known to the model in anyways. If the performance is reported for the same dataset which was used for validation, then I would call it "tricky".

3- Please use Matthews Coorelation Coefficient (MCC) instead of ROC as benchmark measure. ROC can be effected by highly unbalanced dataset. On the other hand MCC is the best measure for unbalanced datasets. It is computed using TP, FP, TN, FN and gives value between -1 and 1.

4- I would suggest you to please make your dataset and codes publicly available for research community, through github or any other website in order to ensure reproducbility and transperancy in research.

I hope the comments are helpful in improving your work.

Author Response

REVIEWER #2

We thank the Reviewer for having provided constructive comments and suggestions.

We have included all suggestions into the revised manuscript.

Comments from writing perspective:

Question 1- Being a reader, I am unable to understand the meaning of Line 110-115. Please elaborate. In particular, line 114-115 feels a bit distracting. If you are not including a certain precursor, why feel the need to put it here. Just go with what you are already doing. 

Answer: We followed the Reviewer’s suggestion and modified the Introduction

Question 2- Line 135-136: I feel these lines need more clear explanation.

Answer: We moved the sentence to the caption of new figure (4). We keep the reference to the bootstrap approach because it is become a kind of standard approach for assessing the uncertainty exploiting the possibility to estimate a given parameters sampling randomly several time the dataset. Being a standard approach we prefer do not include the bootstrap description into the text, which has been already elongated to better explain ML issues.

Question 3- Equation 2: Does this equation mean that sum of moment magnitude of 200 events divided by delta T? If Yes, then perhaps the notation needs changing from Mo to Mo i.

Answer: We thank the Reviewer for having highlighted this typos. We corrected it.

Question 4- Line 218-219: What makes you choose 750 and 2000 around five M4 events?

Answer: We included new text within section 3.2 for explaining the rationale behind our RNNs configuration. We report here for the Reviewer convenience part of the text:

(1) Once the spatial-temporal characteristics (features) of seismicity have been extracted, we selected windows of data for the RNN analyses (i.e., 750 and 2000 earthquakes around the five M4 events for RNN1 and RNN2, respectively). In particular, for each M4 series in RNN1, we consider 499 events before the M4 event, the M4 itself, and 250 events after it. For RNN2, instead, we consider 1500 events before the M4, the M4 itself, and 499 after it. The different amount of data in the M4 series for RNN1 and RNN2 has been tuned for optimizing the training performance.

(3) In order to train the models, we assigned a label to each earthquake. Indeed, being RNN used here for sequential data classification, it is necessary train it with a simple binary (0/1) classification scheme. Therefore, in RNN1, the one aiming to identify the preparatory phase, we assigned value 1 to those events preceding the M4s that have been interpreted based on expert opinion as belonging to the preparatory phase and label 0 to all the others (background and aftershocks). In RNN2, aiming to identify aftershocks, we assigned label 1 to aftershocks and label 0 to the others (background and foreshocks). In particular, in RNN1, for each M4 series, we selected a different number of events as belonging to the preparatory phase (i.e., ranging from 175 to 350 events) looking at the trend of parameters like b-value, Mc and Dc that we know likely related to the preparatory phase [27]. In RNN2, we decided to label all the 499 events following a M4 as aftershocks.

The rationale of our choice is that all features except two (Mw and Dt) are computed on group of events (i.e., each window n = 200). Therefore, the features value at a given instant represents the collective behavior of a group of earthquakes, and it becomes not straightforward to single out, on this basis, if the last event of each window (i.e., the event at ti) is a foreshock, or it belongs to the background seismicity (and similarly for aftershocks and background seismicity). Moreover, these considerations brought us also to avoid considering the location of these events (i.e., latitude, longitude and depth of the earthquakes are not used directly as features, but indirectly through Dc, h and h as collective characteristic of the seismicity), both for RNN1 and RNN2. Our aim during the training phase is to push RNN1 and RNN2 to find coherent correlation among features and to detect changes in the stationary character of the seismicity, which can correspond to the preparatory phase of a large earthquake and to aftershocks, respectively. It is important to note also that our strategy can be a two-edged sword, since the presence of misled background seismicity in both the preparatory and aftershock phases can lead to an underestimation of the performances falsely enhancing the number of missed events. In both cases, we thus make the RNNs work more challenging, and we test its capability to mine and detect coherent correlation among seismic features.

Question 5- Line 226-227: Why switch labelling technique and assign label 1 to aftershocks and 0 to others?

Answer: We use a binary classification scheme in RNN, thus we just use 1 and 0. Algorithms for multiple classifications will be explored in future applications.

Question 6- I feel that there is lack of flow in the manuscript. However, I can be wrong. May I suggest that a clear and separate section or subsection may be created to explain dataset formation. Then a new section or subsection may be made to discuss on RNN application. There should be clear segregation between explanation of dataset and RNN.

Answer: We thank the Reviewer for the suggestion. We modified the work including the new section 3.1 where the methods are explained and we also have included a new figure (Fig. 3) to this purpose.

Question: Since your state of the art is not RNN. Your state of the art is this idea and formation of dataset in this the said manner, thus make a clear section of it.

Answer: We thank the Reviewer for the suggestion. We think that now, having split the method (section 3.1) from our implementation idea (section 3.2), our novel contribution to this research field is more clear.

Comments from analyses perspective:

Question 1- How are you able to identify foreshocks, aftershocks and and background seismicity? Being a student, I think that there are multiple methods for catalog deculstering and their results may vary. Since you do not refer to the method you have used for deculstering or identify foreshocks, aftershocks and background events. It should be referred at least.

Answer: This is an important issue that was raised also by the other Reviewer. We thank the Reviewers for having highlighted it. We included new text to explain the adopted approach in section 3.2.

Please, consider the text reported to answer at Question 4- Line 218-219

Question 2- Line 251-253: It says that Test set is used for Validation. Please ensure that Testing takes place independently of validation. Use different part of dataset for validation. Testing must be done on completely different dataset, which was otherwise not known to the model in anyways. If the performance is reported for the same dataset which was used for validation, then I would call it "tricky".

Answer: We thank the Reviewer for having highlighted this important typo, it was “train” and not “test” indeed. We totally agree with the Reviewer concerns, and in fact we kept separated the training from the testing, using two completely different datasets.

Question 3- Please use Matthews Coorelation Coefficient (MCC) instead of ROC as benchmark measure. ROC can be effected by highly unbalanced dataset. On the other hand MCC is the best measure for unbalanced datasets. It is computed using TP, FP, TN, FN and gives value between -1 and 1.

Answer: We thank the Reviewer for this suggestion. We included MCC in our analysis.

Question 4- I would suggest you to please make your dataset and codes publicly available for research community, through github or any other website in order to ensure reproducbility and transperancy in research.

Answer: This is a good suggestion. The NCEDC seismic catalogue is already freely available. We will organize the features catalogue for its distribution. Concerning the codes, we can of course share them, but it will be made clear that while the method is exportable to other applications, the trained model is not directly applicable to other contexts. This is due mainly to two reasons: the problem and the features are extremely specific for the region, as example, Mc is strictly dependent to the seismic network configuration, while the aftershock productivity depends on the geology; ML techniques cannot well predict input data that lie outside the domain of the train set.

Round 2

Reviewer 1 Report

            Thank you for your close attention to the comments I provided on the previous version of the manuscript. I think the additional details about the method are very helpful and it is much easier to understand what you are doing. I only have a few more minor comments listed below.

ll. 80-113: I think it would be helpful if in the introduction you drew more of a concrete connection between the “preparatory phase” seismicity you discuss in the first ~80 lines of the paper, and the induced seismicity that you are actually analyzing in your study. In particular, it is a little jarring to go from the motivational statement in ll. 80-82, right into the technical description of The Geysers data set in ll. 103-113, without a mention of induced seismicity. One suggestion would be to take the paragraph at ll. 121-124 (and maybe even the following paragraph at ll. 125-129) and move it up to l. 103, to make more of a natural transition from the discussion of the preparatory phase before l. 103, to the more specific case of the induced microseismicity in the Geysers. So, for example, it could read something like this:

“To this aim, we focus on data relevant to The Geysers geothermal field in California. The crustal volume below The Geysers can be seen as a complex physical system whose stress field evolves due to the interaction of tectonic processes and industrial operations in a chaotic way. Our aim is to verify if larger events are anticipated by transients in features sensitive to the evolution of the crustal damage process.

The Geysers hosts a high quality, dense seismic network to monitor the high seismicity rate…..”

l. 404: “(7)” should be “(8)”

l. 409: “(8)” should be “(9)”

Author Response

REVIEWER #1

We thank again the Reviewer for the very useful suggestions he/she provided. We have included into the text the suggested sentence and corrected the typos.

Comments and Suggestions for Authors

Thank you for your close attention to the comments I provided on the previous version of the manuscript. I think the additional details about the method are very helpful and it is much easier to understand what you are doing. I only have a few more minor comments listed below.

  1. 80-113: I think it would be helpful if in the introduction you drew more of a concrete connection between the “preparatory phase” seismicity you discuss in the first ~80 lines of the paper, and the induced seismicity that you are actually analyzing in your study. In particular, it is a little jarring to go from the motivational statement in ll. 80-82, right into the technical description of The Geysers data set in ll. 103-113, without a mention of induced seismicity. One suggestion would be to take the paragraph at ll. 121-124 (and maybe even the following paragraph at ll. 125-129) and move it up to l. 103, to make more of a natural transition from the discussion of the preparatory phase before l. 103, to the more specific case of the induced microseismicity in the Geysers. So, for example, it could read something like this:

“To this aim, we focus on data relevant to The Geysers geothermal field in California. The crustal volume below The Geysers can be seen as a complex physical system whose stress field evolves due to the interaction of tectonic processes and industrial operations in a chaotic way. Our aim is to verify if larger events are anticipated by transients in features sensitive to the evolution of the crustal damage process.

The Geysers hosts a high quality, dense seismic network to monitor the high seismicity rate…..”

Answer: We thank the Reviewer for this suggestion. We think that the suggested sentence for the introduction explaining why The Geysers is interesting for studying the preparatory phase is very useful and we included it into the manuscript.

  1. 404: “(7)” should be “(8)”

Answer: Corrected

  1. 409: “(8)” should be “(9)”

Answer: Corrected

Reviewer 2 Report

I think all my concerns are addressed.

Author Response

REVIEWER #2

As we said, we think that the suggestions provided during first review stage have helped us a lot to improve the manuscript. We thank again the Reviewer.